# Electronic and Molecular Adsorption Properties of Pt-Doped BC_6_N: An Ab-Initio Investigation

**DOI:** 10.3390/nano14090762

**Published:** 2024-04-26

**Authors:** Nada M. Alghamdi, Mohamed M. Fadlallah, Hind M. Al-qahtani, Ahmed A. Maarouf

**Affiliations:** 1Department of Physics, College of Science, Imam Abdulrahman Bin Faisal University, Dammam 31441, Saudi Arabia; nmghamdi@iau.edu.sa; 2Physics Department, Faculty of Science, Benha University, Benha 13518, Egypt; 3Department of Physics, College of Science and Humanities, Imam Abdulrahman Bin Faisal University, Jubail 35811, Saudi Arabia; halqahtani@iau.edu.sa; 4Department of Physics, Faculty of Basic Sciences, German University in Cairo, New Cairo City 11835, Egypt

**Keywords:** hexagonal boron nitride, electronic properties, doping, molecular adsorption, ab-initio calculations

## Abstract

In the last two decades, significant efforts have been particularly invested in two-dimensional (2D) hexagonal boron carbon nitride *h*-B_*x*_C_*y*_N_*z*_ because of its unique physical and chemical characteristics. The presence of the carbon atoms lowers the large gap of its cousin structure, boron nitride (BN), making it more suitable for various applications. Here, we use density functional theory to study the structural, electronic, and magnetic properties of Pt-doped BC_6_N (Pt-BC_6_N, as well as its adsorption potential of small molecular gases (NO, NO_2_, CO_2_, NH_3_). We consider all distinct locations of the Pt atom in the supercell (B, N, and two C sites). Different adsorption locations are also considered for the pristine and Pt-doped systems. The formation energies of all Pt-doped structures are close to those of the pristine system, reflecting their stability. The pristine BC_6_N is semiconducting, so doping with Pt at the B and N sites gives a diluted magnetic semiconductor while doping at the C1 and C2 sites results in a smaller gap semiconductor. We find that all doped structures exhibit direct band gaps. The studied molecules are very weakly physisorbed on the pristine structure. Pt doping leads to much stronger interactions, where NO, NO_2_, and NH_3_ chemisorb on the doped systems, and CO_2_ physiorb, illustrating the doped systems’ potential for gas purification applications. We also find that the adsorption changes the electronic and magnetic properties of the doped systems, inviting their consideration for spintronics and gas sensing.

## 1. Introduction

Since graphene’s fabrication in 2004 [1], two-dimensional (2D) materials have gained considerable experimental and theoretical interest. The reduced dimensionality of these materials and the wide variety of their structural and compositional properties have promised great technological utilization. This interest is boosted by the progress in experimental and characterization methods, which reflects on the potential of fabricating complex structures with specific physical or chemical properties.

Perhaps the most commonly known 2D material is graphene, which possesses unique electronic, mechanical, and optical properties that nominate it for many technological applications [2]. However, some shortcomings in graphene, like its lack of an electronic band gap, led researchers to look for other graphene-like materials [3,4]. Silicene [5], germanene [6], hexagonal boron nitride (h-BN) [7], transition metal dichalcogenides (TMDs) [3,8], MXenes [9,10], and Mo(W)Si_2_N_4_ [11] are examples of 2D materials that mostly have a honeycomb structure, associated with some great features not existing in 3D materials, such as large surface area, high optical transparency, and active sites. The special physical and chemical properties exhibited by 2D materials make them excellent candidates for many potential applications, such as photocatalysis [12], field-effect transistors [13], hydrogen storage [14,15], and gas sensing [16,17,18].

Boron, nitrogen, and carbon are contemporaries on the periodic table. Carbon forms strong covalent bonds with boron and nitrogen, while boron and nitrogen form covalent-polarized bonds. This leads to a variety of (*h*-B_*x*_C_*y*_N_*z*_) compounds [19,20] that recently gained much interest [21]. The replacement of B or N by C may offer a way to tune the properties of those systems without causing any significant lattice distortion. Introducing BN patches into the graphene lattice has been shown to change it from a semimetal to a semiconductor. The Dirac point disappears, and a direct band gap opens at the K points [22,23,24]. It is thus expected that changing the relative compositions of B, C, and N in *h*-B_*x*_C_*y*_N_*z*_ structures may lead to a rich spectrum of physical properties.

Several ternary B-C-N molecule compositions (for example, BCN, BC_2_N, BC_4_N, BC_6_N, and B_2_CN) have been fabricated utilizing a variety of methodologies such as chemical vapor coating, a solvothermal approach, and a chemical reaction approach [25]. BC_2_N also captured the attention of researchers since it is expected to be more thermally stable and chemically inert than diamond and harder than c-BN [22,26]. BCN has a wide range of applications, such as oxidation of contaminants and colorants, hydrogen production from water, clear photovoltaic cell fabrication, UV absorption, optoelectronics, fire-resistant materials, and catalysis for a wide range of chemical interactions [27,28].

Recently, BC_6_N 2D quantum dots have been successfully synthesized by doping graphene quantum dots with boron and nitrogen atoms [29]. The material is found to be a semiconductor with a band gap of 1.2–1.3 eV [30,31]. Because of the graphene-like structure of BC_6_N, it shares some of graphene’s physical and mechanical properties, such as high stiffness and high thermal conductivity [19,31]. Numerous studies have been conducted on BC_6_N to tailor its properties for various applications. It is possible to induce metallic, half-metallic, or dilute magnetic semiconductor behavior by substitutional and adsorption doping in BC_6_N [30,31]. Furthermore, defects such as vacancies alter the electrostatic landscape of BC_6_N, making it sensitive to specific gases [30,31].

With their high intrinsic activity, noble metals can modify the electronic structure of 2D materials and enhance the interaction between gases and nanomaterials. Thus, researchers are becoming more and more interested in noble metal dopings. For example, Pt-doped 2D MoSe_2_ [32,33] can be an efficient electrocatalyst for both hydrogen evolution and oxygen reduction reactions. Pt dopant improves the absorption of 2D Ti_3_C_2_T_2_ Mxene for SF_6_ [34]. CO, NO, NO_2_, O_2_, and NH_3_ molecules can chemisorb on a Pt-doped arsenene sheet [35].

Although pristine BC_6_N has been studied, structures doped with various elements must be investigated. Here, we use spin-polarized density functional theory to analyze the electronic and molecular adsorption properties of Pt-doped BC_6_N. We consider a 3 × 3 supercell of BC_6_N (72 atoms) and study the substitutional Pt-doping of these systems. This supercell allows for a reasonable and experimentally common doping percentage of Pt (∼1%) and minimizes the interaction between adsorbed molecules in neighboring cells, thereby giving a more accurate estimate of the adsorption energies. The supercell has four symmetrically inequivalent sites: B, N, and two C atoms (C1 and C2). We perform the structural relaxation and calculate the densities of states of each system. We then study the adsorption of four gases (NO, CO_2_, NO_2_, and NH_3_) on each system, determining their adsorption energies, as well as their effect on the electronic properties of the doped-BC_6_N nanosheets.

## 2. Computational Methods

We used first-principle calculations based on spin-polarized density functional theory (DFT) as implemented in the Quantum Espresso (QE) plane waves package [36] V6.5 widely used to study the physical properties of periodic systems. The exchange–correlation interaction is described by the generalized gradient approximation (GGA) via the Perdew–Burke–Ernzerhof (PBE) functional [37], and with ultrasoft pseudopotentials (rrkjus-psl version of the QE pseudopotential PS library). An energy cut-off of 55 Ry for the wavefunction, and 550 Ry for the charge density are utilized for all SCF calculations. Our base system is a 3 × 3 supercell of BC_6_N, with a total of *n* = 72 atoms. Doped systems are constructed by substituting a dopant atom for B, N, or C (2 positions). A vacuum spacing of 20 Å avoids any interaction between neighboring images. All systems are fully relaxed (volume and ionic positions) until the forces on the atoms become less than 0.001 Ry/Bohr. Van der Waals interactions (vdW-DF) [38] are included in our study. Brillouin zone integrations are performed using the tetrahedron method and a 12 × 12 × 1 *k*-point grid to calculate the density of states (DOS). Band structures are calculated with 80 *k*-points along the path ΓMKΓ. Charge transfer from/to the BC_6_N systems is determined by calculating the Löwdin charges. The formation energy $E_{f}$ per atom is calculated to check the structural stability of different Pt-BC_6_N systems
(1)$$E_{f}=\left(E\left(\mathrm{Pt}-{\mathrm{BC}}_{6}N\right)+E\left(X\right)-E\left({\mathrm{BC}}_{6}N\right)-E\left(\mathrm{Pt}\right)\right)/n,$$
where $E(\mathrm{Pt}-{\mathrm{BC}}_{6}N)$, E(X), $E({\mathrm{BC}}_{6}N)$, and E(Pt) are the energies of the doped sheet, isolated native atom (B, N, C1 or C2), pristine sheet, and isolated Pt atom in the same supercell, respectively. The adsorption energy is evaluated by
(2)$$E_{ad}=E\left(\mathrm{sheet}\right)+E\left(\mathrm{molecule}\right)-E\left(\mathrm{sheet}+\mathrm{molecule}\right).$$
where E(sheet+molecule), E(sheet), and E(molecule) are the energies of the sheet or doped sheet with the adsorbed molecule, sheet or doped sheet without the adsorbed molecule, and the isolated molecule in the same supercell volume, respectively.

## 3. Results

BC_6_N has a unit cell composed of 6 C atoms, 1 B, and 1 N atoms. The 6 C positions include two distinct sites: a C atom with a B or an N nearest neighbors. In order to study doping with regular experimental percentages, we construct a 3 × 3 supercell of BC_6_N with a total of 72 atoms. To establish a reference, we begin by calculating the electronic properties of this pristine system. The optimized atomic structure of a hexagonal BC_6_N is shown in Figure 1a. Our calculated bond lengths are: C-C = 1.41 Å, C-B = 1.47 Å, and C-N = 1.46 Å, which is in agreement with previous work [30,31,39,40].

Figure 1b shows the projected density of states of the pristine system. The system is semiconducting with a bandgap of 1.3 eV, which matches previous work [30,31,39,40]. The main contributions to the valence and conduction bands are due to the C states, with a small contribution from the N(B) states—the second dominant species in the valence (conduction) band. The bandgap of BC_6_N means that it can be a promising material for optoelectronic applications. The Löwdin charge analysis shows that C1 and N gain electronic charge, while C2 and B lose some charge, consistent with the electronegativities of those atoms and their neighbors.

We now begin to dope our systems. A look at the BC_6_N original unit cell (black parallelogram, Figure 1a shows that there are four distinct locations. Two locations are at the B and N atoms, and the other two are the sites of the C atoms in black circles. The two carbon sites are inequivalent. One site (C1) is connected to two carbon atoms and one boron atom, while the other (C2) is connected to two carbon atoms and one nitrogen site.

### 3.1. Pt-Doping

The relaxed structures of the Pt-doped systems are shown in Figure 2a–d. Minimum deformation in the vicinity of the Pt atom occurs with the Pt at the B site (Pt_B_), the C1 site (Pt_C1_), and the C2 site (Pt_C2_), (Figure 2a–c), and the Pt atom is the out-of-plane bulge of about 1.80 Å with bond lengths (with the nearby atoms) of 1.97 Å, 2.08 Å, 1.96 Å, respectively. The bond length of Pt_C2_ is large due to the atomic size of the B atom as compared to the C and N atoms. The other case is Pt instead of N (Pt_N_, which exhibits an out-of-plane bulge of about 1.6 Å and bond length of 1.97 Å (Figure 2d). The vertical deformation can be understood with the help of the system’s Löwdin charge analysis. For the Pt_N_ doped system, the charge difference ΔQ on the Pt atom and its three nearest neighbors is +0.39*e*, −0.17*e*, −0.167*e* and −0.168*e*. The corresponding distribution of the Pt_C2_ doped system is +0.43*e*, −0.15*e*, −0.042*e*, and −0.005*e*. The bulge around the Pt atom is caused by the difference between the Pt charge. and the charges of the neighboring atoms cause the bulge. A much smaller out-of-plane deformation is seen for the Pt_B_-doped system, with +0.08*e*, +0.05*e*, +0.06*e* and +0.07*e*, and for the Pt_C1_-doped system, with +0.1*e*, −0.08*e*, −0.1*e* and −0.16*e*. The formation energy calculations (Equation (Equation 1)) indicate the Pt_N_-BC_6_N has a slightly lower energy (0.05 eV) as compared to the other doped structures, which have the same formation energy per atom (0.09 eV), Table 1.

The DOS spectrum of the Pt_B_-BC_6_N is spin asymmetric (Figure 2e), especially in the conduction band, which sees significant contributions from the Pt states (such as at 0.2 eV and 0.4 eV) and the C states. The valence band continues to be dominated by the C states. The band gap of the spin-down spectrum is similar to that of the pristine system (1.3 eV), while the spin-up gap becomes 0.2 eV due to the occupation of states localized on the Pt atom and its three nearest carbon neighbors. The asymmetric behavior of the two spin components indicates the structure is magnetic with 0.9 $\mu_{B}$, and hence, the structure is a diluted magnetic semiconductor (DMSC). For Pt_C1_- and Pt_C2_-BC_6_N, the DOS of the two spin channels is symmetric, so the structure is non-magnetic. Due to the contribution of Pt states, as in the valence band, the band gaps decrease to 1.1 eV and 0.5 eV, compared to pristine BC_6_N. The last substitutional doping is Pt_N_. The Fermi energy is shifted to the conduction band, and the structure is a diluted semiconductor with a spin-up and spin-down band gap of 0.2 eV and 1.4 eV, respectively. The asymmetric behavior of the DOS for two spin components in the conduction band gives the structure a magnetization of 1 $\mu_{B}$. All band gaps are direct ones, as is clear from the band structures of the Pt-doped systems (Figure 2i–l).

### 3.2. Molecules Adsorption on Pristine and Pt-Doped BC_6_N Systems

We now discuss the molecular adsorption properties of our systems. We consider four gas molecules, NO, NO_2_, CO_2_, and NH_3_. Each molecule is initially placed above each of the distinct sites (TB (@_TB_), TN (@_TN_), H1 (@_H1_), and H2 (@_H2_)) for pristine BC_6_N, and (Pt_B_, Pt_C1_, Pt_C2_, and Pt_N_) for the doped Pt-BC_6_N sheet, and then the system is structurally relaxed. The relaxation starts from a high symmetry orientation of the molecule so that we allow it to choose the atom it faces the sheet with. We then calculate the system’s various electronic parameters. We will include the adsorption results on pristine structures to form a reference for comparison.

#### 3.2.1. Adsorption of NO

The molecular N-O bond length is 1.16 Å, which agrees well with previous calculations [9]. When NO is placed on the pristine sheet above any position, there is no significant change in its bond length. We find the distances between the N atom and the nearest atoms of our 4 sheets are 2.83 Å, 3.10 Å, 3.10 Å, and 2.88 Å, and with adsorption energies of 0.15 eV, 0.14 eV, 0.12 eV, and 0.17 eV, for the TB, TN, H1, and H2 positions, respectively (Table 2). The NO molecules adsorb very weakly (*E_ad_* < 0.2 eV) on the pristine structure at all adsorption sites (Table 2), leading to negligible effects on the DOS of the pristine system (Figure 3). As expected, no significant charge transfer occurs (Table 2).

The situation is very different with the Pt-doped system. The closest distances between the N atom of the molecule and the Pt dopant atom are 1.93 Å, 1.90 Å, 1.87 Å, and 1.98 Å for NO@Pt_TB_-, NO@Pt_TN_-, NO@Pt_C1_-, and NO@Pt_C2_-BC_6_N, respectively. These are much smaller than the corresponding distances for the pristine system, indicating the much stronger interaction between the NO molecule and the doped sheet. Furthermore, the bond length of N-O increases to 1.198 Å for NO@Pt_B_, NO@Pt_N_ and 1.2 Å for NO@Pt_C1_, NO@Pt_C2_. The adsorption energies are significantly higher: 2.65 eV, 2.27 eV, 2.27 eV, and 2.06 eV for Pt_B_-, Pt_C1_-, Pt_C2_-, and Pt_N_-BC_6_N sheets.

Figure 4e shows the DOS/PDOS of NO@Pt_B_-BC_6_N. The top of the valence band is disturbed by the N and O states, such as for the spin-up component. Also, the mid-gap states are created at 0.4 eV by the N, O, and C states. The band gap becomes 1.1 eV (0.8 eV) for the spin-up (spin-down) component. The structure is a DMSC as Pt_B_-BC_6_N, with a magnetic moment of 2 $\mu_{B}$.

For adsorption atop the Pt at the C1 site, the valence and conduction bands of the structure are disturbed asymmetrically for the two spin components, leading to a DMSC with a magnetic moment of 1 $\mu_{B}$ (Figure 4f). The band gap of the spin-up/down component is 0.9 eV. The P_C2_-BC_6_N structure (Figure 4g) is a half metal, with a spin-down band gap of 0.2 eV and magnetic moment of 1 $\mu_{B}$, which can be utilized for spintronic applications. Finally, adsorption on the Pt_N_-BC_6_N system exhibits the highest structural deformation at the Pt_N_ site (Figure 4h), leading to a higher effect on the DOS compared to adsorption at other locations. The system has a magnetization of 2 $\mu_{B}$ and spin-up/down gaps of 1.2 eV/0.9 eV. Charge analysis and charge density maps (Figure 4i–l) indicate that charge is transferred from the sheet to the NO molecule, causing the polarization of the N atom of the gas to decrease, and that of the O atom to increase.

#### 3.2.2. Adsorption of NO_2_

We now discuss our NO_2_ adsorption results (Figure 5). For the isolated molecule, the N-O bond length and the O-N-O angle are 1.21 Å and 134° [9]. Adsorption on the pristine sheet slightly changes the structure of the molecule; the bond length and the angle slightly change to 1.23 Å and 127° for adsorption on TB, TN, and H1, while the bond angle is about 128.6° for the H2 position. The molecule is very weakly adsorbed at all sites (0.1 eV), and it faces the sheet with an O atom at a distance of ∼2.92 Å for the TB, TN, and H1 sites (Table 3). At the H2 site, it is a bit closer (2.8 Å), with its N atom facing the sheet. A small amount of charge is transferred from the sheet to the molecule. Because of the weak adsorption of NO_2_ on the pristine system, its effect on the electronic properties of the sheet is minimal, but for the sake of completion, those results are reported in the Appendix A.

We now discuss our results for the adsorption of NO_2_ on the Pt-doped nanosheets. The N-O bond length is ∼1.21 Å for all systems. However, the O-N-O angle decreases to 112.3°, 123.3°, 124.8° and 111.8° for Pt_B_-, Pt_C1_-, Pt_C2_- and Pt_N_-BC_6_N, respectively. The change in the angle, such as for the cases of NO_2_@Pt_B_- and NO_2_@Pt_N_-BC_6_N reflects the strong interaction between the nitrogen dioxide molecule and the doped sheet. This is further confirmed by the shorter distance between the molecule and the Pt atom, as well as the charge transfer between the molecule and the sheets (∼0.35*e*, Table 3). The charge density maps (Figure 5i–l) show the distribution of the transferred charge around the NO_2_ molecule Therefore, the adsorption of NO_2_ is improved by Pt doping. As expected, the NO_2_@Pt_B_- and NO_2_@Pt_N_-BC_6_N have higher adsorption energy as compared to the other positions in the doped monolayer because of the maximal charge localization occurring at the B and N sites.

In Figure 5e–h, we show the effect of NO_2_ on the electronic properties of the sheets. Adsorption changes the state and the magnetic moment of Pt-BC_6_N, from DMSC to SC, for NO_2_@Pt_N_ with a band gap of 1.3 eV, from SC to metal for NO_2_@Pt_C1_ and NO_2_@Pt_C2_-BC_6_N, and from DMSC to SC with a band gap of 1.23 eV for NO_2_@Pt_B_-BC_6_N. All the NO_2_@Pt-BC_6_N monolayers are nonmagnetic, except NO_2_@Pt_C1_-BC_6_N, which shows a small magnetization (0.2 $\mu_{B}$).

#### 3.2.3. Adsorption of CO_2_

CO_2_ is the second triatomic molecule that we studied. At all considered locations, the average bond length and the O-C-O angle are nearly unchanged (1.17 Å, 179.3° [9]). The interaction with the sheet is weak (0.2 eV, see Table 4 and Appendix A), which is reflected in the equilibrium distance from the sheet (∼3.2 Å), and the negligible charge transfer with the sheet (Table 4).

The picture with the Pt-doped system is quite different. The doped sheet adsorbs the CO_2_ molecule at a distance of ∼2.2 Å, and the O-C-O angle decreases significantly. The electronic structures of CO_2_@Pt_B_- and CO_2_@Pt_N_-BC_6_N are very similar. They are DMSCs with a magnetic moment of 1.0 $\mu_{B}$ ((Figure 6e,h), and with spin-up (down) band gaps of 1.0 eV (0.8 eV), and 1.0 eV (0.9 eV), for CO_2_@Pt_B_- and CO_2_@Pt_N_-BC_6_N, respectively. On the other hand, CO_2_@Pt_C1_- and Pt_C2_-BC_6_N are non-magnetic semiconductors, with a band gap of 1.1 eV and 0.4 eV, respectively, for both spin directions (Figure 6f,g). The charge transfer from the sheet to the molecule is higher than with the pristine system, and its distribution is shown in Figure 6i–l.

#### 3.2.4. Adsorption of NH_3_

The last molecule in our study is NH_3_. On the pristine system, the bond length and H-N-H angle of the molecule (1.02 Å and 106.67° [9]) do not change, except for NH_3_@_TB_BC_6_N, where the angle is 102.2° (Appendix A). The optimized distance between the closest H atom of the molecule and the sheet is 2.7 Å (Table 5). There is very little charge transfer (<0.1*e*). Those values reflect the weakness of the interaction between the molecule and the pristine sheet. Indeed, we calculate an adsorption energy of ∼0.2 eV.

As with the other studied molecules, the Pt-doped system greatly enhances the adsorption of NH_3_ (Table 5). Also, the electronic properties become those of a DMSC with a spin-up (down) band gap of 1.3 eV (0.2 eV) for NH_3_@Pt_B_-BC_6_N (Figure 7e–h). NH_3_@Pt_C1_-BC_6_N and NH_3_@Pt_C2_-BC_6_N are semiconductors with spin-up (down) band gaps of 1.1 eV and 0.7 eV. Finally, NH_3_@Pt_N_-BC_6_N is a half-metal. Our charge analysis indicates that, contrary to three previous molecules, NH_3_ loses charge to the sheet in the four configurations. This is also illustrated in the charge density maps of Figure 7i–l.

Previous research has demonstrated that Pt-doped 2D materials indeed enhance molecular adsorption and sensing [41]. For example, CO and NO are found to chemisorb on Pt-doped MoS_2_ monolayers with adsorption energies of 1.38 eV and 1.21 eV, respectively [42]. Pt-doped monolayer WSe_2_ is shown to be suitable for the adsorption of CO_2_, NO_2_, and SO_2_ [43]. A monolayer of Pt-doped HfSe_2_ exhibits desirable adsorption behavior for SO_2_ and SOF_2_ [44]. Our results discussed are in line with previous findings. Computing the average adsorption energy of each gas on the different adsorption sites, one can assess the potential of the Pt-doped BC_6_N for gas filtration. As shown in Table 6, NO_2_, NO, and NH_3_ are chemisorbed on the Pt-doped systems, while CO_2_ is physisorbed. This suggests that our Pt-doped systems are suitable for the filtration of NO_2_, NO, and NH_3_, and to a lesser degree for CO_2_. We have also seen that the adsorption of NO_2_, NO, NH_3_, and CO_2_ leads to some changes in the band gaps of our Pt-doped systems, which may be utilized to develop sensors for those gases. Therefore, the Pt-doped BC_6_N systems studied in this work are good candidate materials for the filtration/sensing of NO_2_, NO, NH_3_, and CO_2_.

## 4. Conclusions

In this work, we use first principle calculations to investigate the structural, electronic, and magnetic properties of Pt-doped BC_6_N (Pt-BC_6_N). We also study the adsorption of four common gas molecules (NO, NO_2_, CO_2_, and NH_3_) on the pristine and doped structures. We consider the substitutional doping occurring at the four distinct sites in the BC_6_N lattice: the B, the N, and two distinct C sites. The formation energies of the doped structures indicate that they are stable. Upon Pt-doping, the BC_6_N acquires a magnetization of ∼1 $\mu_{B}$ for doping at the B and N sites, whereas systems doped at the two distinct C sites remain non-magnetic. The gaps of all doped systems change due to the Pt-doped states that are now located in the pristine gap. We also study the molecular adsorption properties of the doped structures using four gases: NO, NO_2_, CO_2_, and NH_3_, where the gas molecule is placed close to the Pt atom. Adsorption of NO, NO_2_, CO_2_, and NH_3_ is significantly enhanced on the doped structures. NO, NO_2_, and NH_3_ chemisorb with energies of ∼1.3–3 eV, while CO_2_ physisorb with 0.7 eV. The interaction between the gas molecules and the doped sheets results in some charge transfer. This turns the doped systems non-magnetic for NO_2_, CO_2_, and NH_3_, while it increases the magnetic moment for NO. The results of our study are important for applications of BC_6_N, which include spintronics, gas filtration, and molecular sensing.

## Figures and Tables

**Figure 1 nanomaterials-14-00762-f001:**
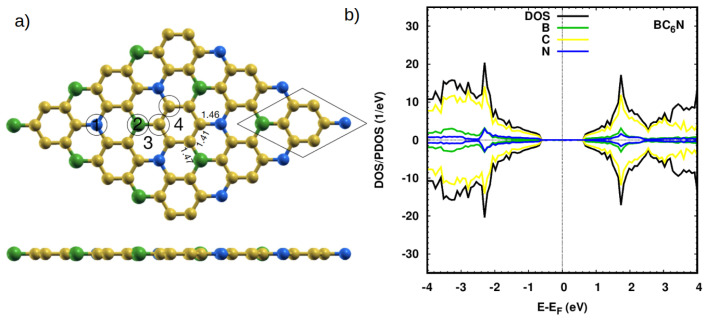
(**a**) Top and side view of the optimized atomic structure of a pristine hexagonal 3 × 3 supercell of a BC_6_N sheet. The black parallelogram marks the original unit cell of BC_6_N. Yellow, green, and blue spheres represent C, B, and N atoms, respectively. Bond lengths are given in Å. The numbers “1” (top on N (TN)), “2” (top on B (TB)), “3” (top on BNC hexagonal (H1)), and “4” (top on C hexagonal (H2)) indicate the adsorption sites of gases on the BC_6_N, whereas the circles refer to the doping sites where the systems are doped with a Pt atom. (**b**) Projected DOS (PDOS) of the pristine BC_6_N system showing the most significant contributions.

**Figure 2 nanomaterials-14-00762-f002:**
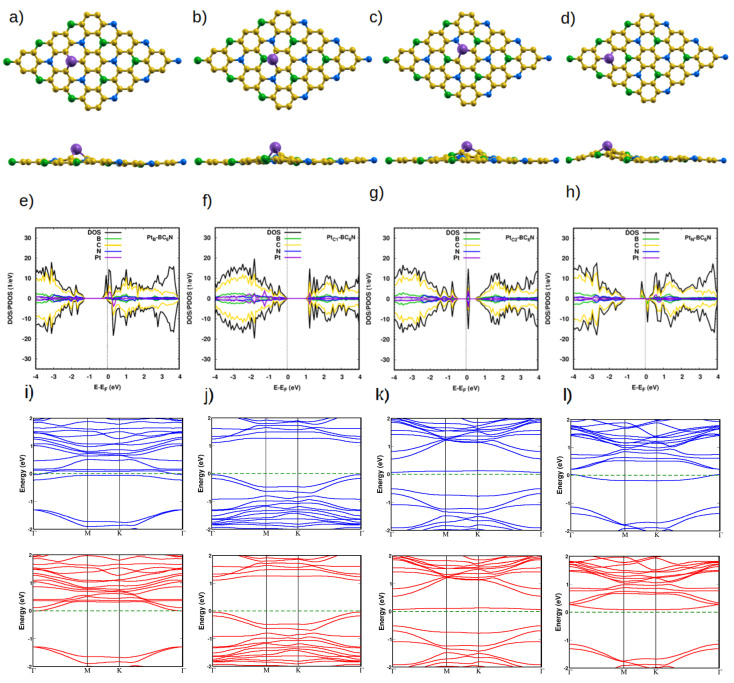
(**a**–**d**) Top and side view of the optimized atomic structure of Pt-BC_6_N sheet with different doping sites, Pt_B_, Pt_C1_, Pt_C2_, and Pt_N_, respectively. The purple sphere represents the Pt atom. (**e**–**h**) The corresponding DOS/projected DOS (PDOS) of the Pt-BC_6_N system shows the most significant contributions. (**i**–**l**) The corresponding band structure, from −2 eV to 2 eV. Spin-up (down) is in blue (red).

**Figure 3 nanomaterials-14-00762-f003:**
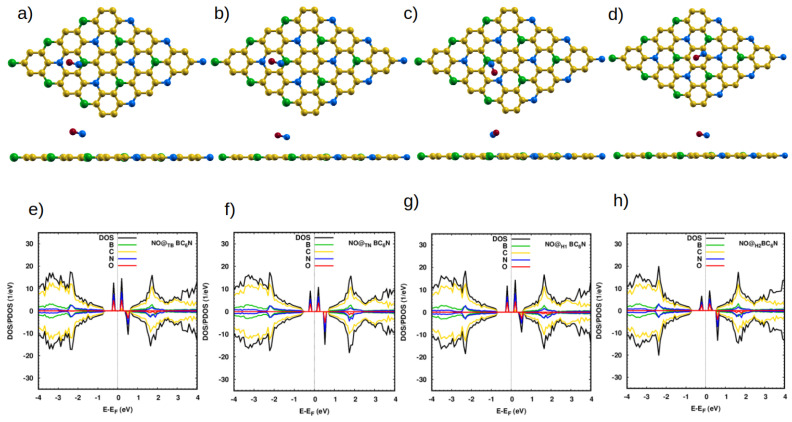
Top and side view of the optimized atomic structure of NO@BC_6_N sheet with different doping sites: (**a**) NO@_B_BC_6_N, (**b**) NO@_N_BC_6_N, (**c**) NO@_H1_BC_6_N and (**d**) NO@_H2_BC_6_N. (**e**–**h**) The corresponding DOS/projected DOS (PDOS) of the NO@BC_6_N systems. The red spheres represent O atoms.

**Figure 4 nanomaterials-14-00762-f004:**
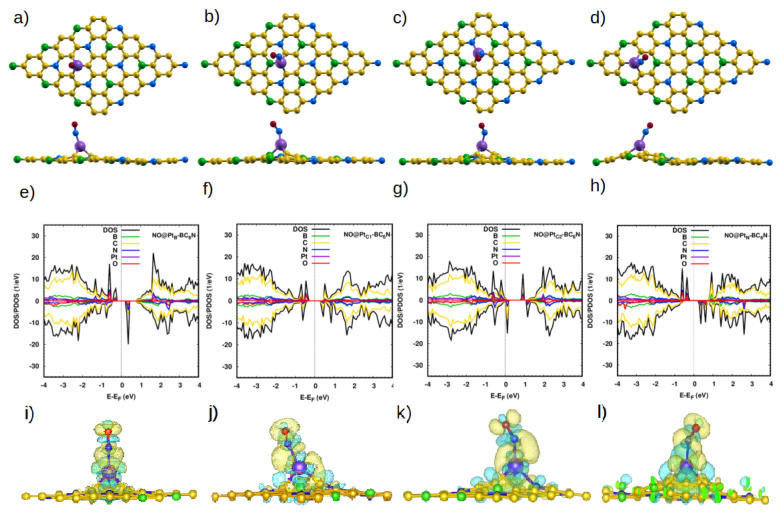
(**a**–**d**) Top and side view of the optimized atomic structure of NO@Pt-BC_6_N sheet with different doping sites: (**a**) NO@Pt_B_-BC_6_N, (**b**) NO@Pt_C1_-BC_6_N, (**c**) NO@Pt_C2_-BC_6_N, and (**d**) NO@Pt_N_-BC_6_N. (**e**–**h**) The corresponding DOS/projected DOS (PDOS) and (**i**–**l**) the corresponding charge densities of the NO@Pt-BC_6_N systems. Isosurface yellow (blue) color represents higher (lower) charge density.

**Figure 5 nanomaterials-14-00762-f005:**
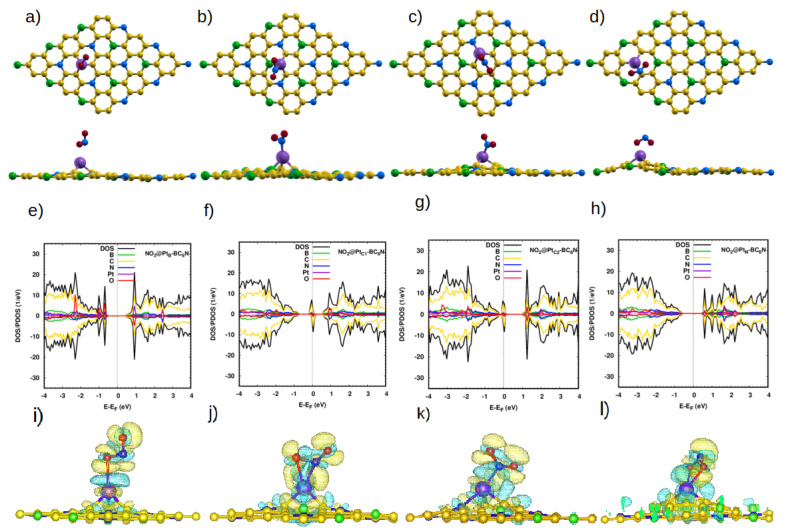
(**a**–**d**) Top and side view of the optimized atomic structure of NO_2_@Pt-BC_6_N sheet with different doping sites: (**a**) NO_2_@Pt_B_-BC_6_N, (**b**) NO_2_@Pt_C1_-BC_6_N, (**c**) NO_2_@Pt_C2_-BC_6_N, and (**d**) NO_2_@Pt_N_-BC_6_N. (**e**–**h**) The corresponding DOS/projected DOS (PDOS) and (**i**–**l**) the corresponding charge densities of the NO_2_@Pt-BC_6_N systems. Isosurface yellow (blue) color represents higher (lower) charge density.

**Figure 6 nanomaterials-14-00762-f006:**
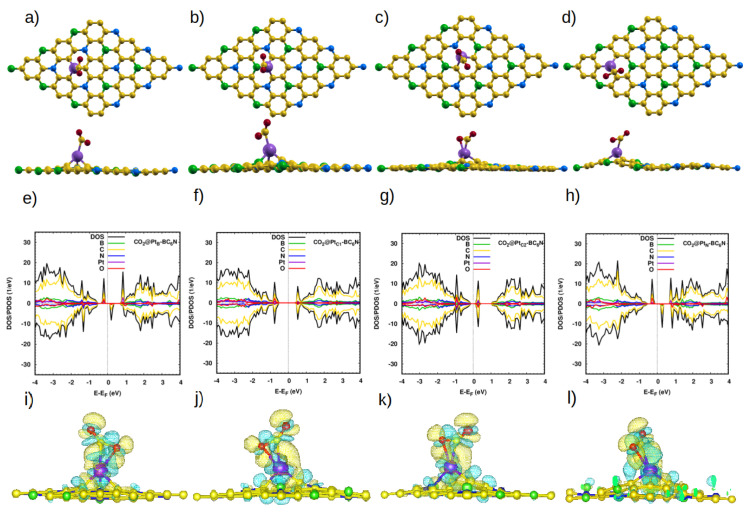
(**a**–**d**) Top and side view of the optimized atomic structure of CO_2_@Pt-BC_6_N sheet with different doping sites: (**a**) CO_2_@Pt_B_-BC_6_N, (**b**) CO_2_@Pt_C1_-BC_6_N, (**c**) CO_2_@Pt_C2_-BC_6_N, and (**d**) CO_2_@Pt_N_-BC_6_N. (**e**–**h**) The corresponding DOS/projected DOS (PDOS) of the CO_2_@Pt-BC_6_N systems. (**i**–**l**) The corresponding charge density. Isosurface yellow (blue) color represents higher (lower) charge density.

**Figure 7 nanomaterials-14-00762-f007:**
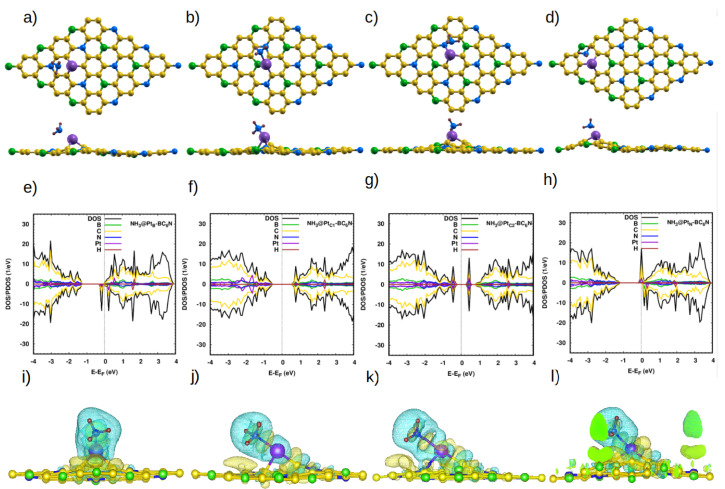
(**a**–**d**) Top and side view of the optimized atomic structure of NH_3_@Pt-BC_6_N sheet with different doping sites: (**a**) NH_3_@Pt_B_-BC_6_N, (**b**) NH_3_@Pt_C1_-BC_6_N, (**c**) NH_3_@Pt_C2_-BC_6_N, and (**d**) NH_3_@Pt_N_-BC_6_N. (**e**–**h**) The corresponding DOS/projected DOS (PDOS) and (**i**–**l**) the corresponding charge densities of the NH_3_@Pt-BC_6_N systems. Isosurface yellow (blue) color represents higher (lower) charge density. In (**l**), the isosurface extends to the supercell border, and the green color indicates the interior of the isosurface.

**Table 1 nanomaterials-14-00762-t001:** Pt-BC_6_N: the charge transfer (ΔQ (*e*)), formation energy ($E_{f}$ (meV)), magnetization (Mag ($\mu_{B}$)), band gap (*E_g_* (eV)).

Systems	ΔQ	$E_{f}$	Mag	$E_{g}$ (up)	$E_{g}$ (dn)
BC_6_N	-	-	0.0	1.3	1.3
Pt_B_-BC_6_N	0.08	87	0.9	0.2	1.3
Pt_C1_-BC_6_N	0.10	88	0.0	1.1	1.1
Pt_C2_-BC_6_N	0.43	94	0.0	0.5	0.5
Pt_N_-BC_6_N	0.39	51	1.0	0.2	1.4

**Table 2 nanomaterials-14-00762-t002:** NO adsorption NO pristine and Pt-BC_6_N: the closest distance (*d* (Å)), nearest atom (X), charge transfer (ΔQ (*e*)), adsorption energy ($E_{ad}$ (eV)), magnetization (Mag ($\mu_{B}$)), band gap ($E_{g}$ (eV)).

Systems	d	X	ΔQ	$E_{\mathrm{ad}}$	Mag	$E_{g}$ (up)	$E_{g}$ (dn)
@_TB_BC_6_N	2.8	N	−0.04	0.2	1.0	0.4	1.1
@_TN_BC_6_N	3.1	N	0.00	0.1	1.0	1.1	0.4
@_H1_BC_6_N	3.1	N	−0.04	0.1	1.0	0.3	1.2
@_H2_BC_6_N	2.9	N	−0.05	0.2	1.0	0.5	1.3
@Pt_B_-BC_6_N	1.9	N	−0.20	2.7	2.1	1.1	0.8
@Pt_C1_-BC_6_N	1.9	N	−0.21	2.3	1.0	0.9	0.9
@Pt_C2_-BC_6_N	1.9	N	−0.15	2.3	0.7	0.2	-
@Pt_N_-BC_6_N	2.0	N	−0.20	2.1	2.1	1.2	0.9

**Table 3 nanomaterials-14-00762-t003:** NO_2_ adsorption on pristine and PtBC_6_N: the closest distance (*d* (Å)), O-N-O angle ($\theta^{\circ }$), nearest atom (X), charge transfer (ΔQ (*e*)), adsorption energy ($E_{ad}$ (eV)), magnetization (Mag ($\mu_{B}$)), and band gap ($E_{g}$ (up) and down (dn) (eV)).

Systems	d	$\theta^{\circ }$	X	ΔQ	$E_{\mathrm{ad}}$	Mag	$E_{g}$ (up)	$E_{g}$ (dn)
@_TB_BC_6_N	2.9	126.7	O	−0.07	0.1	0.9	1.4	-
@_TN_BC_6_N	3.0	126.9	O	−0.16	0.1	0.9	1.4	-
@_H1_BC_6_N	3.0	126.9	O	−0.08	0.1	−0.9	-	1.3
@_H2_BC_6_N	2.8	128.6	N	−0.11	0.1	0.9	0.3	1.3
@Pt_B_-BC_6_N	2.0	112.3	O	−0.43	3.0	0.0	1.3	1.3
@Pt_C1_-BC_6_N	2.1	123.3	N	−0.34	2.5	0.2	-	-
@Pt_C2_-BC_6_N	2.0	124.8	N	−0.30	2.5	0.0	-	-
@Pt_N_-BC_6_N	2.3	111.8	O	−0.41	3.3	0.0	1.2	1.2

**Table 4 nanomaterials-14-00762-t004:** CO_2_ adsorption on pristine and Pt-BC_6_N: The closest distance (*d* (Å)), O-C-O angle ($\theta^{\circ }$), nearest atom (X), charge transfer (ΔQ (*e*)), adsorption energy ($E_{ad}$ (eV)), magnetization (Mag ($\mu_{B}$)), and band gap ($E_{g}$ (up) and down (dn) (eV)).

Systems	d	$\theta^{\circ }$	X	ΔQ	$E_{\mathrm{ad}}$	Mag	$E_{g}$ (up)	$E_{g}$ (dn)
@_TB_BC_6_N	3.3	179.3	C	−0.02	0.2	0.0	1.3	1.3
@_TN_BC_6_N	3.2	179.8	C	−0.01	0.2	0.0	1.3	1.3
@_H1_BC_6_N	3.2	179.3	C	−0.02	0.2	0.0	1.3	1.3
@_H2_BC_6_N	3.2	179.3	C	−0.02	0.2	0.0	1.3	1.3
@Pt_B_-BC_6_N	2.2	144.0	C	−0.38	0.7	1.0	1.0	0.8
@Pt_C1_-BC_6_N	2.1	141.5	C	−0.42	0.8	0.0	1.1	1.1
@Pt_C2_-BC_6_N	2.1	141.0	C	−0.42	0.6	0.0	0.4	0.4
@Pt_N_-BC_6_N	2.2	146.4	C	−0.33	0.4	1.0	1.0	0.9

**Table 5 nanomaterials-14-00762-t005:** NH_3_ adsorption on pristine and Pt-BC_6_N: the closest distance (*d* (Å)), H-N-H angle ($\theta^{\circ }$), nearest atom (X), charge transfer (ΔQ (*e*)), adsorption energy ($E_{ad}$ (eV)), magnetization (Mag ($\mu_{B}$)), and band gap ($E_{g}$ (up) and down (dn) (eV)).

Systems	d	$\theta^{\circ }$	X	ΔQ	$E_{\mathrm{ad}}$	Mag	$E_{g}$ (up)	$E_{g}$ (dn)
@_TB_BC_6_N	2.7	102.2	H	−0.01	0.2	0.0	1.3	1.3
@_TN_BC_6_N	2.7	106.4	H	−0.01	0.2	0.0	1.3	1.3
@_H1_BC_6_N	2.7	106.2	H	−0.01	0.2	0.0	1.3	1.3
@_H2_BC_6_N	2.8	106.2	H	−0.01	0.2	0.0	1.3	1.3
@Pt_B_-BC_6_N	2.2	107.3	N	0.22	1.5	−0.9	1.3	0.2
@Pt_C1_-BC_6_N	2.2	107.1	N	0.23	1.6	0.0	1.1	1.1
@Pt_C2_-BC_6_N	2.3	107.3	N	0.21	1.3	0.0	0.7	0.7
@Pt_N_-BC_6_N	2.3	106.9	N	0.22	1.2	0.9	-	1.4

**Table 6 nanomaterials-14-00762-t006:** Average adsorption energies ($E_{ad}^{ave}$ (eV)) on the Pt-doped systems. The average is taken over the 4 different adsorption sites in each system.

Gas	$E_{\mathrm{ad}}^{\mathrm{ave}}$
NO_2_	2.8
NO	2.4
NH_3_	1.4
CO_2_	0.6

## Data Availability

The data presented in this study are available in article.

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
