# Peer review of "Electronic and Molecular Adsorption Properties of Pt-Doped BC6N: An Ab-Initio Investigation"

_nanomaterials, 2024, doi:10.3390/nano14090762_

Round 1
Reviewer 1 Report
Comments and Suggestions for Authors
The manuscript “Electronic and Molecular Absorption Properties of Pt-doped BC6N: An Ab-initio Investigation”, the authors presented the theoretical calculation of carbon doped hBN and molecular gases absorption such as NO, NO2, CO2, NH3.
Recently, carbon defects in hBN have emerged as promising materials for quantum emitters due to their stable formation energy. So, theoretical studies of carbon-related BN structure and their doping effect by metal and molecular gases are interesting topics.
Theoretical studies of several carbon defects in hBN, such as Song Li et al., J. Phys. Chem. Lett, 13, 3150 (2022) have reported that robust 4.1 eV energy states formation inside the bandgap of hBN. The BC6N configuration has low formation energy. These theoretical studies are also supported by experiment results of 4.1 eV CL and PL from carbon-doped hBN.
In the manuscript, the authors claimed that BC6N possesses the 1.3 eV bandgap, which is inconsistent with the reported results of 4.1 eV. The authors must discuss distinct features for 4.1 eV carbon defects in hBN and their atomic configurations.
Furthermore, the authors presented several molecular gas absorption and electronic properties in the manuscript. However, it needs to be made clear which configuration is stable for each gas absorption and their properties. Thus, it is recommended to summarize these to enhance readers' understanding.
The manuscript needs major revision before publication to Nanomaterials.
Author Response
Reviewer 1
"Theoretical studies of several carbon defects in hBN, such as Song Li et al., J. Phys. Chem. Lett, 13, 3150 (2022) have reported that robust 4.1 eV energy states formation inside the bandgap of hBN. The BC6N configuration has low formation energy. These theoretical studies are also supported by experiment results of 4.1 eV CL and PL from carbon-doped hBN.
In the manuscript, the authors claimed that BC6N possesses the 1.3 eV bandgap, which is inconsistent with the reported results of 4.1 eV. The authors must discuss distinct features for 4.1 eV carbon defects in hBN and their atomic configurations."
The reviewer correctly pointed out that the 4.1 eV gap is for the pristine h-BN. Our base system, being majorly doped with carbon, is BC6N. Our 1.3 eV calculated gap of BC6N indeed matches very closely what has been published in the literature for BC6N. Please see the following references:
1- Band gap = 1.228 eV
Aghaei, S.; Aasi, A.; Farhangdoust, S.; Panchapakesan, B. Graphene-like BC6N nanosheets are potential candidates for detection of volatile organic compounds (VOCs) in human breath: A DFT study. Applied Surface Science 2021, 536, 147756.
2- Band gap = 1.3 eV
Bafekry, A. Graphene-like BC6N single-layer: Tunable electronic and magnetic properties via thickness, gating, topological defects, and adatom/molecule. Physica E: Low-dimensional Systems and Nanostructures 2020, 118, 113850.
3- Band gap of 1.27 eV.
Babar, V.; Sharma, S.; Schwingenschlogl, U. Gas sensing performance of pristine and monovacant C6BN monolayers evaluated by density functional theory and the nonequilibrium green’s function formalism. The Journal of Physical Chemistry C 2020, 124, 5853–5860.2
4- Band gap = 1.28 eV
Aasi, A.; Mehdi Aghaei, S.; Panchapakesan, B. Outstanding performance of transition-metal-decorated single-layer graphene-like BC6N nanosheets for disease biomarker detection in human breath. ACS omega 2021, 6, 4696–4707.
"Furthermore, the authors presented several molecular gas absorption and electronic properties in the manuscript. However, it needs to be made clear which configuration is stable for each gas absorption and their properties. Thus, it is recommended to summarize these to enhance readers' understanding. "
If the reviewer refers to the stability of the molecule-sheet structure at room temperature, then judging by the magnitude of the adsorption energies (~ 0.4 - 3.3 eV), which are all much bigger than the room temp ( kT = 0.025 eV), we see that all molecule-sheet structures are stable at room temperature. However, if the reviewer refers to the stability issue amongst the different molecules, then per his/her recommendation, we have added a table summarizing the adsorption results at the end of the discussion in the revised manuscript.
Reviewer 2 Report
Comments and Suggestions for Authors
In the work, the authors used the density functional theory, to simulate the electronic and molecular adsorption properties of two-dimensional materials, Pt-doped BC6N. Before the publication, the manuscript should be revised, as the following:
1. In line 11, “a smaller gap” is direct or indirect?
2. In line 38, “strong bonds” are covalent or partially-covalent?
3. Details of GGA should be described.
Comments on the Quality of English LanguageThe scientific English should be concise. However, some sentences are too long. For example, in the line 37-39, “Being contemporaries on the periodic table, boron, nitrogen, and carbon can form 37 strong bonds, leading to a variety of (h-BxCyNz) compounds [23,24], which have gained 38 much interest [25], as the replacing of boron or nitrogen by carbon does not cause any 39 significant distortion in the lattice.” They should be improved.
Author Response
Reviewer 2
"In the work, the authors used the density functional theory, to simulate the electronic and molecular adsorption properties of two-dimensional materials, Pt-doped BC6N. Before the publication, the manuscript should be revised, as the following:
1. In line 11, “a smaller gap” is direct or indirect?”
To address the reviewer’s question, we calculated all the band structures of the doped systems, and added them to the revised manuscript. All band gaps are direct.
“2. In line 38, “strong bonds” are covalent or partially-covalent?”
The reviewer asks about the nature of the bonding in carbon-boron-nitrogen structures. The C-N and C-B bonds are covalent, whereas the B-N bonds are covalent-polarized bonds (please see:
Mierzwa G, Gordon AJ, Berski S. The nature of multiple boron-nitrogen bonds studied using electron localization function (ELF), electron density (AIM), and natural bond orbital (NBO) methods. J Mol Model. 2020 May 13;26(6):136. doi: 10.1007/s00894-020-04374-9. PMID: 32405959; PMCID: PMC7220893.)
“3. Details of GGA should be described. “
We have added more details about the parameters of our GGA calculations in the computational details section of the revised manuscript.
" The scientific English should be concise. However, some sentences are too long. For example, in the line 37-39, “Being contemporaries on the periodic table, boron, nitrogen, and carbon can form 37 strong bonds, leading to a variety of (h-BxCyNz) compounds [23,24], which have gained 38 much interest [25], as the replacing of boron or nitrogen by carbon does not cause any 39 significant distortion in the lattice.” They should be improved."
We acknowledge the lengthiness of the sentence. We have modified it to address the reviewer’s point. Furthermore, other long sentences have been made more clear.
Reviewer 3 Report
Comments and Suggestions for Authors
1. In section 1. Introduction, it is recommended to discuss the methods and software programs used for Ab-initio Investigation. To justify the choice of DFT and Quantum Espresso package for the conducted research.
2. In Section 3. Results, justify the choice of the calculation model (Figure 1). For example, why 3×3 supercell is selected? How various possible structural defects will affect the results? It is recommended to add files with models for calculation in additional materials.
3. To decipher the abbreviations: DOS, DMSC.
4. It is recommended to discuss the results obtained. What is the scientific significance of the results obtained? Were there any innovations in the Ab-initio Investigation? To make a comparison with similar calculations for other 2D materials. How can the results obtained be applied in the engineering practice of designing various devices?
5. How are you planning to continue this research? For example, conducting an experimental validation of the results obtained.
Author Response
Reviewer 3
“1. In section 1. Introduction, it is recommended to discuss the methods and software programs used for Ab-initio Investigation. To justify the choice of DFT and Quantum Espresso package for the conducted research.”
We value the recommendation of the reviewer. Quantum Espresso is the most widely used DFT packages used to study fundamental properties materials. A justification of that is not typically seen in DFT papers, but to follow the reviewer’s recommendation we eluded to that in the computational details section.
“2. In Section 3. Results, justify the choice of the calculation model (Figure 1). For example, why 3×3 supercell is selected? How various possible structural defects will affect the results? It is recommended to add files with models for calculation in additional materials.”
The 3x3 supercell is selected to allow for reasonable and experimentally common doping percentage of Pt (~ 1%). Furthermore, the 3x3 supercell minimizes the interaction between adsorbed gas molecules in neighboring cells, thereby giving a more accurate estimate of the adsorption energies. We have added these justifying arguments to the revised manuscript.
“3. To decipher the abbreviations: DOS, DMSC."
We have added them to the revised manuscript.
“4. It is recommended to discuss the results obtained. What is the scientific significance of the results obtained? Were there any innovations in the Ab-initio Investigation? To make a comparison with similar calculations for other 2D materials. How can the results obtained be applied in the engineering practice of designing various devices?”
The results obtained show the potential of the studied systems for use in gas filtration devices. In the revised manuscript, we have added a paragraph following the recommendation of the reviewer on the potential use of the studied Pt-doped BC6N systems as materials for gas filtration.
“5. How are you planning to continue this research? For example, conducting an experimental validation of the results obtained.”
Currently, we are working on an extension of this project where we study the doping of the BC6N systems using other elements for applications in molecular adsorption, including hydrogen storage. Experimental validation is beyond the scope of our expertise, but our results may indeed encourage experimental physicists to prepare and investigate the studied systems.
Reviewer 4 Report
Comments and Suggestions for Authors
The manuscript, entitled " Electronic and Molecular Adsorption Properties of Pt-doped BC6N: An Ab-initio Investigation", presented a systematical investigation for adsorption behaviors of small molecular gases (NO,NO2, CO2, NH3) on Pt-doped BC6N by ab-initio calculation. The results have been well discussed and are potentially helpful for gas purification applications. The manuscript is recommended for acception after making up some additional informations about the charge redistribution.
The charge redistribution, i.e., charge density and difference, for pristine and adsorped gas molecules system, should be presented for vividly comparing the charge transfer among the system.
Author Response
Reviewer 4
“The charge redistribution, i.e., charge density and difference, for pristine and adsorped gas molecules system, should be presented for vividly comparing the charge transfer among the system.”
We value the reviewer's request. We have calculated the charge density difference maps of all studied molecule-sheet systems and added them to the revised manuscript.